# Quantum Dot Photoluminescence Enhancement in GaAs Nanopillar Oligomers Driven by Collective Magnetic Modes

**DOI:** 10.3390/nano13030507

**Published:** 2023-01-27

**Authors:** Maria K. Kroychuk, Alexander S. Shorokhov, Damir F. Yagudin, Maxim V. Rakhlin, Grigorii V. Klimko, Alexey A. Toropov, Tatiana V. Shubina, Andrey A. Fedyanin

**Affiliations:** 1Faculty of Physics, Lomonosov Moscow State University, 119991 Moscow, Russia; 2Ioffe Institute, 194021 St. Petersburg, Russia

**Keywords:** dielectric oligomers, quantum dots, absorption efficiency enhancement, emission enhancement

## Abstract

Single photon sources based on semiconductor quantum dots are one of the most prospective elements for optical quantum computing and cryptography. Such systems are often based on Bragg resonators, which provide several ways to control the emission of quantum dots. However, the fabrication of periodic structures with many thin layers is difficult. On the other hand, the coupling of single-photon sources with resonant nanoclusters made of high-index dielectric materials is known as a promising way for emission control. Our experiments and calculations show that the excitation of magnetic Mie-type resonance by linearly polarized light in a GaAs nanopillar oligomer with embedded InAs quantum dots leads to quantum emitters absorption efficiency enhancement. Moreover, the nanoresonator at the wavelength of magnetic dipole resonance also acts as a nanoantenna for a generated signal, allowing control over its radiation spatial profile. We experimentally demonstrated an order of magnitude emission enhancement and numerically reached forty times gain in comparison with unstructured film. These findings highlight the potential of quantum dots coupling with Mie-resonant oligomers collective modes for nanoscale single-photon sources development.

## 1. Introduction

Development of quantum information science, including encoding, transmission, and processing of information, as well as related technologies, such as quantum cryptography, key distribution, and quantum communication boosted creation of nanoscale sources of single photons [1,2,3]. Many attempts have been made to realize single-photon sources at single atoms [4], ions [5] and molecules [6] and on various color centers [7]. However, from the commercial point of view, the other promising candidates are semiconductor quantum dots (QDs) [8,9]—tens-of-nanometers-sized particles of smaller-band-gap semiconductor embedded in a larger-band-gap material, based on (In, Ga, Al)As. Such structures possess small emission linewidth, fast radiative decay time, high and stable quantum efficiency compared to other single-photon sources [10]. They also demonstrated the ability to create emitters for the near-IR communication ranges (see [11,12] and references therein). The growth technology of semiconductor QDs [13] is well established, and such systems are easily scalable [14] and integrable with existing devices [8,15,16].

A typical way to increase the rate of single-photon QDs emission is integrating them into either dielectric or metallic resonators. Dielectric systems can support resonator modes with a significant quality factor (Q-factor), which leads to a noticeable increase in radiation due to the Purcell effect in the weak coupling regime [17,18,19,20,21]. The most popular resonators are based on Bragg structures (often columnar) and planar photonic crystals. Recent progress in these area has made it possible to create bright single photon emitters with a high degree of indistinguishability and purity [22,23,24]. However, their use requires a precise balance between the radiation rate and the quality factor [25], as well as solving the problem of radiation extraction [26]. In addition, it is difficult to create both a multilayer Bragg structure using epitaxial methods and a photonic crystal with a periodic array of holes using electron beam lithography. Alternative coupling to surface plasmons in metallic structures [27,28,29,30,31] suffers from high optical losses [32], which leads to both limiting the emission intensity of QDs and damaging the potential devices by heating them. Interaction of QDs with various semiconductor [33,34,35,36,37,38] and plasmonic waveguides [39,40] also allows one to control their photoluminescence (PL) by coupling to either propagating waveguide modes or surface plasmon–polariton modes. However, these approaches suffer the same limitations as described above.

The coupling of QDs to lossless all-dielectric nanoparticles with optical response governed by multipolar Mie-type resonances [41] gained increasing popularity over the last decade because their fabrication is markedly easier. Such nanostructures have proved their effectiveness in linear [42] and nonlinear [43] light manipulation at the nanoscale. For efficient interaction of the dipole sources with Mie-resonant nanoparticles, they should be localized close to [44] or inside [45,46,47] the nanostructure. Different location of quantum source, shape and material of nanoobjects affect the photonic mode density [48] resulting in PL intensity variations via Purcell effect [49]. They also act as nanoantennas [50] changing initially symmetric emission pattern of QDs, especially under the Kerker conditions [51,52], thus increasing the number of single photons reaching the detectors. On the other hand, the presence of all-dielectric nanoantennas [53,54] influences the number of photons absorbed by QDs and thus their PL intensity due to the strong local field confinement in their volume, especially at the wavelength of magnetic dipolar Mie-type resonance (MDR) [55]. Strong near-field interaction may occur between all-dielectric nanoparticles with subwavelength spacings. Combining them in isolated nanoclusters—oligomers—leads to an excitation of collective modes [56], which manifest themselves both in linear [57] and nonlinear [58,59] optical responses of the system. In particular, for the Ge(Si) QDs coupled with collective Mie modes of an array of silicon nanopillars, the Q-factor of 500 has been demonstrated [60]. The MDR of oligomers is characterized by strong local field confinement inside nanoparticles essential for increasing quantum sources emission intensity.

In this work, we experimentally and numerically investigate the interaction of InAs QDs with MDR of isolated oligomers—quadrumers of GaAs nanopillars—grown on the GaAs substrate (see Figure 1). By exciting such clusters with linearly polarized light (LP), we gain considerable intensity boosting of individual emission lines of QDs. In particular, we experimentally show more than ten times QDs PL enhancement compared to unstructured film when the excitation wavelength is in the spectral vicinity of MDR. We also define the mechanism for their emission engineering and propose the nanoresonator geometry suitable for independent control over QDs excitation and luminescence.

## 2. Materials and Methods

### 2.1. Sample Fabrication

The experiments were carried out using the sample fabricated by molecular beam epitaxy, which contained self-assembled InAs QDs grown using the Stranski–Krastanov growth mode. The intended average thickness of the InAs deposition amounted to 1.9 monolayers (MLs). The structure was grown on a GaAs (100) substrate capped with a 0.2 µm thick GaAs buffer, followed by InAs QDs formed between the 200 nm-thick bottom and 200 nm-thick top GaAs barrier layers at the growth temperature of 590 °C and the growth rate of 0.015 ML/s; see Appendix A. After the growth, the negative resist was applied to the structure by centrifugation, and then the mask was exposed by electron beam lithography for the further patterning. The last step was etching through the resulting mask to the depth of the grown GaAs layer by the reactive ion etching. Parameters of the mask and the etching depth were chosen according to numerical optimization of structure scattering cross-sections performed using the finite element method in COMSOL multiphysics.

### 2.2. Dark-Field Spectroscopy

Scattering cross-sections of isolated quadrumers with embedded QDs were measured on a custom-built setup for dark-field spectroscopy in reflection configuration (schematically presented in Figure 2a) under LP illumination. A combination of the halogen lamp with lenses (L1) being part of the lamp (Thorlabs - SLS201L/M), the multi-mode optical fiber, Thorlabs FT600EMT, (F1), and the collimator (C), doublet Thorlabs AC254-60-B-ML, was used as a broadband source in the experiment. The collimated light beam passed through the lens (L2), doublet Thorlabs AC254-150-B-M, and the objective lens (O) Lomo 20× with NA = 0.7 confocal with the lens L2, and illuminated several oligomers in the experimental sample mounted on the three-coordinate translator, placed in the focal plane of the objective lens (O). Radiation reflected from the substrate and scattered from the quadrumers passed through O and confocal lenses L2 and L3, doublet Thorlabs AC254-150-B-M, with the equal focal length of 150 mm. Since contribution of the light reflected from the substrate prevailed over the scattered light from oligomers, back-focal plane spatial filtering was applied for separation of the reflected and scattered radiation in k-space that was in the back focal plane of the O. However, the back focal plane of the O was inside its body and, for its visualization, L2 and L3 were used while the real quadrumers images were projected to the camera by the set of lenses L4. By moving the opaque mask between L3 and L4, we discovered the back focal plane image and blocked zero diffraction order with homemade mask in the form of a spot of paint (d≈1 mm) on a piece of glass (CS), thus extracting reflected light. Depending on the position of the mirror on the flip mount (FM), a dark-field image of the sample was projected either to the CMOS-camera (Thorlabs, DCC1545M) or to the facet of the multi-mode optical fiber, Thorlabs FT600EMT, (F2) connected to the spectrometer (Ocean Optics QE Pro). Varying the diameter of the field aperture (iris diaphragm SM1D12), we selected the signal of the individual oligomer and measured its scattering cross-section as (Signal–Background)/(Lamp–Noise) with the spectrometer with the 2-nm spectral resolution. The noise corresponded to the effects associated with the dark current of the spectrometer. The background was measured on the sample area without particles to avoid parasitic scattering from substrate irregularities and optical elements. To obtain the reference spectrum of the lamp, we put the mirror instead of the sample and removed the back-focal mask out of the setup. The resulting scattering cross-sections were the arithmetic mean of at least five spectra for each oligomer.

### 2.3. Micro-PL Spectroscopy

Micro-PL setup shown schematically in Figure 3a was used to investigate QDs PL enhancement under excitation by LP radiation at cryogenic temperature of 8 K. The samples were mounted in a He-flow cryostat ST-500-Attocube with the XYZ piezo-driver inside, which allowed one to optimize and precisely maintain the positioning of the chosen quadrumer. Optical excitation by spectrally tunable Coherent Mira 900 laser system working in continuous-wave operation regime was used for the µ-PL measurements. The spectral range of excitation was chosen according to emission features of InAs QDs and MDR positions of oligomers. The incident radiation was focused on the sample’s 3–5 µm spot by an apoachromatic objective lens (O) with NA of 0.42. PL radiation reflected from the structure was focused by a triplet achromatic lens (L1) in the plane of the mirror with the calibration aperture (PH) of various diameters (minimum available aperture size is 50 µm) that specified the area where the PL signal was measured. PL intensity transmitted through the aperture was collected and focused on the entrance slit of the spectrometer SP-2500 (Princeton Instruments) with a 1200 mm grating by the set of two triplet lenses (L2). A set of shortpass and longpass Semrock filters (F) was placed just before the spectrometer for blocking the parasitic reflection of the pump radiation from the sample and optical elements.

## 3. Results and Discussion

Efficiency of detected QDs emission depends on the interplay of two effects. The first one is the QDs interaction with quadrumers at the PL wavelength, which increases QDs emission directivity. This means the growth of the number of photons captured by an objective lens. The second effect concerns the number of pump photons absorbed by QDs that is sensitive to the electric field localization factor in the volume of nanopillars, L(λ), depending on the type of optical mode in the oligomer at the excitation wavelength. In this work, both factors were studied. We excited MDR of the quadrumer with high L(λ) by LP radiation resulting in enhanced absorption of pumping photons by the QDs and increased QDs emission efficiency. It was also possible to pick specific geometric parameters of the oligomer for MDR overlapping with the emission spectrum of QDs. This resulted in additional enhancement of measured PL intensity due to the improved directivity.

Two sets of structures were considered. The first one was attributed to the case when the MDR spectral position located outside of the QD PL spectral lines, which, according to μ-PL measurements, corresponded approximately to the range from 890 nm to 990 nm. The second one was designed to achieve the best possible spectral overlap between MDR and QD emission spectra. Isolated oligomers of four GaAs nanopillars were obtained (Figure 2b inset) with fixed height of h=400 nm, spacing between nanoparticles of s=100 nm, and various diameters of d=210:290 nm. The 10 μm distance between oligomers was selected to exclude interaction between neighboring nanoclusters and make it possible to excite each structure separately.

Scattering cross-section spectra of two representative quadrumers are presented in Figure 2b. Spectra of the other quadrumers can be found in the Appendix A, together with typical dark-field images of quadrumers and monomers shown in Appendix A. The spectra showed the resonant behavior in the spectral range between 650 nm and 950 nm. Wavelengths larger than 900 nm were not visualized due to the setup limitations. The resonance was red shifted with the increase in diameter of nanopillars. Numerical simulations using the finite-difference time-domain (FDTD) method in Lumerical FDTD software were performed to identify the origin of the observed resonances. The calculated scattering cross-section spectrum of the quadrumer, shown in Appendix A, reveals the peak corresponding to the magnetic dipole mode of the GaAs nanopillars. The local electric field distribution at the wavelength of 925 nm in the inset of Appendix A displayed the MDR profile in high-index nanoparticles. Numerical calculations of the scattering cross sections of quadrumers for different diameters were also carried out using the finite element method (see Appendix A) in COMSOL Multiphysics, which also confirmed excitation of the MDR.

First we characterized unpatterned GaAs films with embedded InAs QDs by μ-PL at 8 K with 404 nm excitation wavelength. The signal was collected from the area of 2–3 μm. The PL spectrum is shown in Figure 3b, revealing two separate broad peaks attributed to the QDs emission in the spectral range from 890 nm to 1000 nm and emission of the wetting layer from 850 nm to 880 nm. As the structure was patterned in isolated quadrumers, the InAs QDs emission spectrum was changed. The μ-PL spectrum revealed several relatively narrow lines (Figure 3b, black curve) centered between 920 nm and 970 nm. For additional characterization of QDs, we measured their PL spectra for various pump powers and fixed excitation laser wavelength, see Appendix A. The peak in the spectra observed for low intensities was associated with emission from the first excited level of a quantum dot. Another peak corresponding to the emission from the next excited level is observed for the increased pump power.

Then, we studied the emission of QDs localized in quadrumers of different diameters (whose scattering cross-section spectra are shown in Figure 2b). The pump wavelength was tuned from 690 nm to 819 nm outside the resonant excitation range of QDs. We measured the intensities of several individual emission lines that were stable during the experiment. Raw emission spectra are presented in Appendix A. The emission lines whose intensities for different pump wavelengths are shown in Figure 3b are also indicated. After obtaining the entire spectrum, the procedure was repeated three times. The resulting spectra were the latter’s mean normalized to the maximum value in the spectrum. Yellow dots in Figure 3c represented the case when the oligomer’s MDR was in the spectral vicinity of QDs PL excitation wavelength, and the emission lines were located out of the resonances of oligomers. The MDR of the nonresonant quadrumer was shifted from the excitation wavelength but was still off-resonant for QDs (red dots). Five times enhancement of PL intensity was achieved for QDs embedded in resonant quadrumers. The signal in the nonresonant case showed no valuable sensitivity to the change of the excitation wavelength.

Mie-resonant oligomers also acted as nanoantennas for QDs PL radiation. Varying the nanopillar diameter, one could tune the MDR spectral position and reach the case when the QDs emission wavelength coincides with the Mie-type mode. We showed experimentally that measured PL intensity was increased at least three times for the excitation in the over-barrier regime as the MDR spectral position approached the QDs PL wavelength (Figure 3d). The QDs emission spacial profile lost the axial symmetry typical for the source in an unstructured film (see Appendix A). PL was strongly modified by the radiation pattern of the MDR of the quadrumer. It remained practically symmetrical for nonresonant samples. The enhancement values of the detected PL intensity for both factors are presented in Appendix A.

We also studied QDs PL intensity enhancement by numerical modeling in COMSOL Multiphysics software. The results are shown in Figure 4. QDs inside each nanopillar were represented by the point electric dipoles placed at the particles’ mid-height. QDs’ spatial distribution inside the nanopillar is discussed in Appendix A. The dipole moment of the point source embedded into the nanoresonator was proportional to the electric field localized inside the particle at the pump wavelength, $d\left(\lambda_{pump}\right)∼E_{loc}\left(\lambda_{pump}\right)$, modulated by the Lorentz function with the maximum at 950 nm according to the results of the micro-PL analysis of InAs QDs. Dipoles acted as the sources of electromagnetic radiation that was mainly coupled to the Mie-resonant modes of the quadrumer efficiently excited at the emission wavelength. The PL amplification had the resonance when the source turned out to be at the field maximum of the mode, and the orientation of the dipole coincided with the polarization of the electric field. The radiation power of dipoles was integrated over the sphere around the resonator and the value $P_{dip}\left(\lambda_{pump},\lambda_{PL}\right)$ obtained was qualitatively characterized the QDs emission intensity. To highlight the impact of oligomers to PL intensity, we normalized $P_{dip}\left(\lambda_{pump},\lambda_{PL}\right)$ to the radiation power of dipoles placed inside the unstructured film.

Using the described above method, we simultaneously considered the interaction of QDs emission with nanopillar and amplification of QDs absorption due to the presence of quadrumer collective resonances by varying the dipole source amplitude. The PL intensity reached 40-times enhancement in the spectral vicinity of MDR (Figure 4a). The highest value of PL intensity was observed as the pump wavelength became equal to $\lambda_{PL}=900$ nm that was slightly shifted from the MDR because the local fields responsible for the signal gain were not necessarily at their maximum in the vicinity of the scattering peak maximum (see Appendix A).

PL resonance for LP case was spectrally broad with FWHM of more than 100 nm due to the low quality factor of the MDR of the GaAs quadrumer on a GaAs substrate. It is possible to excite resonances of a quadrumer with a higher quality factor, which was shown in our previous work [61]. We realized the excitation of the collective out-of-plane magnetic mode (OMD) of the quadrumer by azimuthally polarized (AP) beams. OMD was characterized by strong local field confinement in the nanopillar essential for increasing quantum sources’ emission intensity. We numerically showed that it was possible to combine the enhancement of QDs absorption and the PL directivity with the use of AP excitation. We picked such geometric parameters of the oligomer that MDR overlapped with the emission spectrum of QDs while their excitation was independently governed by AP light (Figure 4a). When the AP pump wavelength excited the OMD mode of the quadrumer ($\lambda_{pump}=895$ nm), the QDs PL intensity reached the maximum value at the MDR wavelength that was four times greater (160 times compared to the unstructured film) than for the LP case.

## 4. Conclusions

To summarize, InAs QDs PL enhancement with MDR of GaAs oligomer was experimentally realized. Physical principles underlying the gain were discussed, and numerical calculations were carried out to supplement the achieved results. The hybrid system under study was fabricated and characterized via scanning electron microscopy, optical spectroscopy, and μ-PL. We experimentally demonstrated the increase in the PL intensity of QDs by at least an order of magnitude when they are excited at the wavelength of quadrumer’s MDR. The effect can be associated with the localization of the electric field in the structure increasing the light absorption of QDs at the MDR. The PL intensity enhancement was also experimentally shown by changing the QDs emission pattern by nanopillars. Three times QDs PL enhancement is obtained when the MDR of oligomers overlaps with the QDs emission spectra. Numerical modeling demonstrates forty times PL enhancement when two effects are combined. For further boosting of QDs PL, we proposed using AP pump beams for nanocluster excitation. Additional amplification of QDs emission with AP excitation by four times compared to LP pumping was numerically realized. Such nanophotonic structures can serve as functional elements for applications in applied quantum optics on a chip.

## Figures and Tables

**Figure 1 nanomaterials-13-00507-f001:**
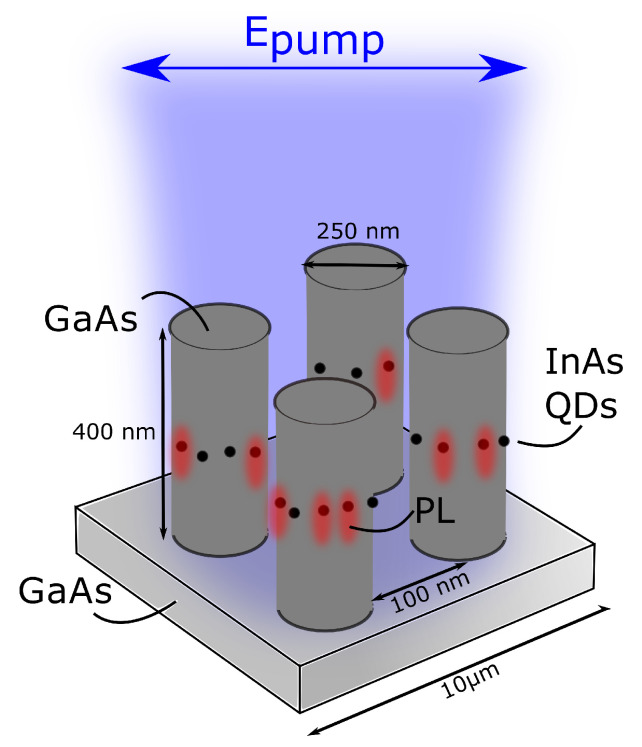
Schematic representation of PL enhancement in InAs QDs embedded into the GaAs quadrumer.

**Figure 2 nanomaterials-13-00507-f002:**
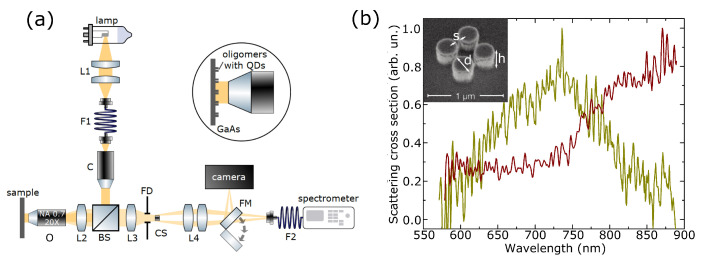
**Optical characterization of the quadrumer.** (**a**) Experimental setup for dark-field spectroscopy measurements of quadrumers with embedded QDs. L1:4 are lenses, O is the objective lens, F1:2 are optical fibers, C is collimator, BS is the beam splitter, FM is the flipper mirror, FD is the field aperture. (**b**) Experimental scattering cross-section spectra of isolated oligomers of nanopillars with diameters of 220 nm (yellow) and 260 nm (red). Inset shows scanning electron microscope image of the sample of Mie-resonant quadrumers coupled with QDs; spacings between nanopillars are s=100±5 nm; diameters are d=250±5 nm; heights are h=400±5 nm.

**Figure 3 nanomaterials-13-00507-f003:**
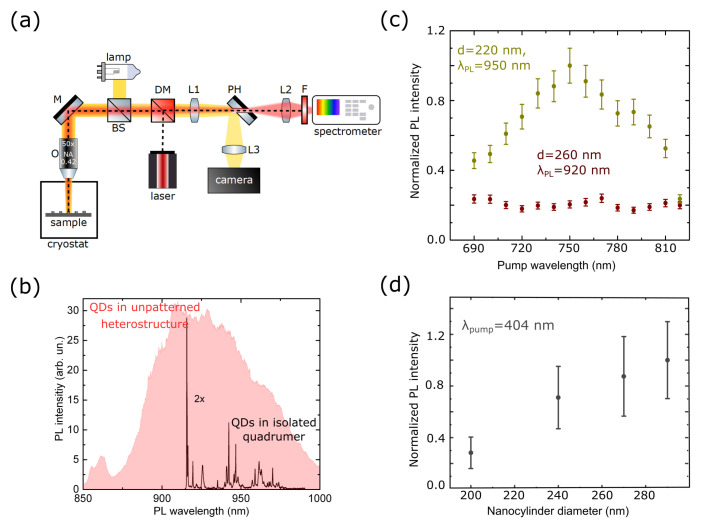
**Microphotoluminescence of QDs coupled to GaAs quadrumers for linearly polarized pump beams.** (**a**) Experimental setup for microphotoluminescence measurements at cryogenic temperature of 8 K. L1:3 are lenses, O is the objective lens, M is the mirror, BS is the beam splitter, DM is the dichroic mirror, PH is the pinhole coupled to the mirror, F is the set of low pass and long pass filters. (**b**) The red area is the μ-PL spectrum of InAs QDs embedded into the GaAs film, obtained from the 2.5 μm spot on the sample pumped by laser radiation with 404 nm wavelength. The black curve with a set of emission lines is the representative spectrum of InAs QDs located in the GaAs nanopillars forming isolated quadrumer. (**c**) Experimental photoluminescence intensity normalized to the maximum value for resonant, d≈220 nm (yellow), and non-resonant, d≈260 nm (red), samples measured for different pump wavelengths and fixed PL wavelength ($\lambda_{PL}$) for each sample ($\lambda_{PL}=950$ nm and $\lambda_{PL}=920$ nm). (**d**) The dependence of the isolated quadrumer PL intensity on the diameter of nanopillars normalized to its maximum value.

**Figure 4 nanomaterials-13-00507-f004:**
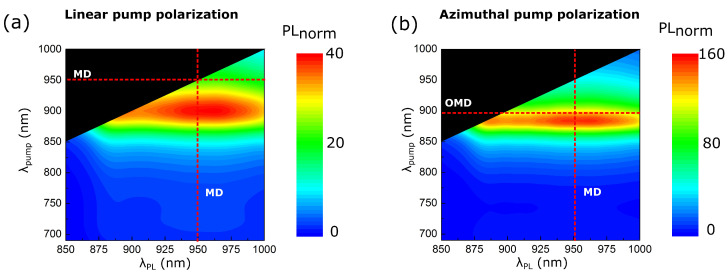
**Numerical results of QDs μ-PL for various pump polarizations.** Numerically obtained PL enhancement for linear (**a**) and azimuthal (**b**) polarizations of the pump radiation for GaAs nanopillars coupled with InAs QDs on the GaAs substrate. MD is the magnetic dipole Mie-type mode, OMD—collective out-of-plane magnetic dipole mode.

## Data Availability

Not applicable.

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
