# Peer review of "Quantum Dot Photoluminescence Enhancement in GaAs Nanopillar Oligomers Driven by Collective Magnetic Modes"

_nanomaterials, 2023, doi:10.3390/nano13030507_

Round 1

Reviewer 1 Report

The work is of interests to the quantum photonics and quantum information research community. The work is overall convincing and worth publishing after following issues/suggestions were addressed/implemented:

1) There are different PL enhancement factors reported for different cases in the paper and readers may lost which number is for which case, my suggestion is to create a table which summarizes the PL enhancement factors

2) Please explain briefly why two different simulation software has been used, COMSOL and Lumerical.

3) In PL simulation as presented in the supplementary, the authors integrated the collection PL signal across the whole top half plane, this is not true since your collection lens NA is only 0.42, which means some signals will not be collected by the objective. Can authors please explain why you have integrated the whole half plane?

4) In Figure 4, what is OMD as in figure b?

5) Supplementary file: wrong reference of fig ??? inside, need correction. Fig. S10, last figure, can you swap topdown the insert image as it doesn't match Figure 1's orientation.

END OF report.

Author Response

We are thankful to all reviewers for careful reading of the manuscript and pointing out the issues and inaccuracies in it.

We are grateful for Reviewer #1 who wrote ‘The work is overall convincing and worth publishing after following issues/suggestions were addressed/implemented’. We also thank him/her for valuable comments and suggestions.

Reviewer 1:

 “There are different PL enhancement factors reported for different cases in the paper and readers may lost which number is for which case, my suggestion is to create a table which summarizes the PL enhancement factors”.

Our response:

We thank the reviewer for his/her comment. We added such table into supplementary materials and provided a link in the main text.

Text added:  (p. 6, paragraph Results and discussion) The enhancement values of the detected PL intensity for both factors are presented in Table 1, Supporting Information.

Reviewer 1:

“Please explain briefly why two different simulation software has been used, COMSOL and Lumerical.”

Our response:

We are grateful for this question. We mainly used Lumerical to calculate the optical response when systems were excited by linear polarization, as the most familiar tool for us. Vector beams, on the other hand, were easier to specify in Comsol, therefore, to analyze photoluminescence for different polarizations, we proceeded with calculations in this software package. We also duplicated the calculation of the scattering cross section of the quadrumer in Comsol (Figure S10 in Supporting Information).

Reviewer 1:

“In PL simulation as presented in the supplementary, the authors integrated the collection PL signal across the whole top half plane, this is not true since your collection lens NA is only 0.42, which means some signals will not be collected by the objective. Can authors please explain why you have integrated the whole half plane?”

Our response:

We thank the reviewer for this valuable comment. To clarify our statement, we performed numerical simulations for the sample upper-half plane (considering everything which can in principle be detected in free space) and then integrated scattering cross-section for different angles (please check Fig. S4(b,e), results for different angles depicted by different colors). We want to emphasize that these results were normalized by the symmetric scattering cross-section of the dipole immersed into the GaAs substrate (200 nm apart from the top surface). Thus, when we consider the whole top half plane we compare overall efficiency of the light outcoupling from the emitter in the non-structured substrate and in the resonant oligomer. PL outcoupling efficiency for emitters in the substrate is limited due to high refractive index contrast between GaAs and air. Resonant oligomers at the same time more effectively outcouple PL radiation into free space, which is demonstrated by the modeling results. However, in the experiment we compared detected PL intensity for resonant and non-resonant oligomers. So, in principle we could divide the results of Fig. S4b and Fig. S4e to compare them with the experimental data, but it’s a little bit tricky since we also need to take into account such factors as dipole orientation and location inside the cluster. That’s why in the Supplementary just for simple illustration we decided to highlight only general PL outcoupling efficiency increase in case of the resonant oligomer in comparison with the non-structured substrate.

Text added:  (Supporting Information, p. 4-5) For each spectral point <...> integration is carried out for the corresponding solid angle. For resonant quadrumer five times overall PL enhancement is observed for π/2 solid angle (Fig. S4(b)) while for the non-resonant oligomer there is no significant increase in the overall PL outcoupling efficiency (Fig. S4(e)).

Reviewer 1:

“In Figure 4, what is OMD as in figure b?

Our response:

It corresponds to the collective out-of-plane magnetic dipole (see line 234 of the manuscript). We agree that it can be confusing since the definition goes after Figure 4 in the text, so we have added more details in the caption.

Text added:  (p. 6, Fig.4) MD is the magnetic dipole Mie-type mode, OMD - collective out-of-plane magnetic dipole mode.

Reviewer 1:

“Supplementary file: wrong reference of fig ??? inside, need correction. Fig. S10, last figure, can you swap topdown the insert image as it doesn't match Figure 1's orientation.”

Our response:

We thank the reviewer for pointing out these issues. We’ve corrected Figure S10 in accord with the reviewer's suggestion. In the new version of Supporting Information it is Figure S9.

Reviewer 2 Report

The submitted article: "Quantum Dot Photoluminescence Enhancement in GaAs Nanopillar Oligomers Driven by Collective Magnetic Modes" fits into the global ground of ongoing research on this topic. The authors presented a very interesting theoretical method, which can then be used to produce a physical element, which the authors also showed. The results obtained are very promising. The article is well-written and enjoyable to read. However, there are many writing errors in the present form e.g. in supplementary material: “Fig. ??(a), nonresonant quadrumer (MDR experience red shift from QDs emission spectrum) Fig. ??(d)”.

The manuscript has sufficient scientific quality and relevance for Nanomaterials. I suggest accepting the manuscript in present form after spelling corrections.

Author Response

We are thankful to all reviewers for careful reading of the manuscript and pointing out the issues and inaccuracies in it.

We thank Reviewer #2 for his/her high rating of our work. We have corrected typos and errors through the text following his/her advise.

Reviewer 2:

 “However, there are many writing errors in the present form e.g. in supplementary material: “Fig. ??(a), nonresonant quadrumer (MDR experience red shift from QDs emission spectrum) Fig. ??(d)””.

Our response:

We thank the reviewer for his/her positive feedback on our work. We have checked the supporting information file and made necessary corrections. They are marked in red in the updated version.

Reviewer 3 Report

The manuscript by Kroychuk et al. is devoted to theoretical and experimental study of quantum-dot photoluminescence (PL) assisted by the GaAs nanopillar oligomers. The effect of PL enhancement due to the plasmonic resonances of metallic nanosctructures is well-known and deeply studied. The analogous phenomenon in dielectric structures is less studied, but it is also possible due to Mie resonances supported by such structures. The authors consider this very possibility investigating PL of quantum dots (QDs) coupled to Mie-resonant oligomers and obtain an order of magnitude emission enhancement. The paper is well-written and contains reliable results. So, I would like to recommend its acceptance.

I have only several minor comments worth to be considered by the authors:

1) I think that the first demonstration of QD PL enhancement near plasmonic structures (Kulakovich et al., 2002, doi: 10.1021/nl025819k) is worth to be cited in the proper place of introduction.

2) It is known that spontaneous emission can be not only enhanced, but also inhibited near the dielectric resonant nanoparticles (see, e.g., Muravitskaya et al., 2022, doi: 10.1021/acs.jpcc.1c09844). Is it possible to observe something similar in the oligomer structure studied in the manuscript?

3) I cannot entirely agree with the interpretation that the improved directivity results in PL enhancement (line 148). PL is enhanced due to two factors: (i) the localization of pumping radiation close to a plasmonic or Mie resonance, (ii) the increased local density of states (LDOS) at the PL frequency. None of these factors deals with directivity. As far as I understand, the directivity per se cannot enhance PL. It can only increase the signal registered in a given direction, but PL (i.e., integral radiation emitted over all directions) will be the same.

4) Check the supplementary information for some inconsistencies ("Fig. ??" on page 3, "other works concerning all-dielectric nanophotonics []" on page 5).

Author Response

We are thankful to all reviewers for careful reading of the manuscript and pointing out the issues and inaccuracies in it.

We thank Reviewer #3 for his/her valuable comment about different factors for photoluminescence enhancement. We agree with the provided statement and have corrected the corresponding part of the manuscript in accord with it.

Reviewer 3:

“I think that the first demonstration of QD PL enhancement near plasmonic structures (Kulakovich et al., 2002, doi: 10.1021/nl025819k) is worth to be cited in the proper place of introduction.

Our response:

We thank the reviewer for his/her advice and add the reference in the introduction when describing coupling with plasmonic systems.

Text added:  (p. 2, main text)  Alternative coupling to surface plasmons in metallic structures [27]

Reviewer 3:

“It is known that spontaneous emission can be not only enhanced, but also inhibited near the dielectric resonant nanoparticles (see, e.g., Muravitskaya et al., 2022, doi: 10.1021/acs.jpcc.1c09844). Is it possible to observe something similar in the oligomer structure studied in the manuscript?

Our response:

Indeed it is possible and we thank the reviewer for this comment. This will depend on the configuration of the optical modes formed in the cluster and also on the position of the QDs inside the nanopillars forming it. We believe that it can be useful, for example, for canceling out unwanted emission lines in the spectra, but it goes out of the scope of this work and can be investigated more thoroughly in the future study.

Text added:  (p. 2,  paragraph 2, main text) Different location of quantum source, shape and material of nanoobjects affect the photonic mode density [48] resulting in PL intensity variations via Purcell effect (either a decrease or an increase depending on the configuration [49]).

Reviewer 3:

“I cannot entirely agree with the interpretation that the improved directivity results in PL enhancement (line 148). PL is enhanced due to two factors: (i) the localization of pumping radiation close to a plasmonic or Mie resonance, (ii) the increased local density of states (LDOS) at the PL frequency. None of these factors deals with directivity. As far as I understand, the directivity per se cannot enhance PL. It can only increase the signal registered in a given direction, but PL (i.e., integral radiation emitted over all directions) will be the same.

Our response:

We thank the reviewer for pointing out this issue. We fully agree with this comment and corrected our statement in the main text to emphasize the enhancement of the measured PL intensity.

Text added:  (p. 4, paragraph Results and discussion) Efficiency of detected QDs <...>. This resulted in additional enhancement of measured PL intensity <...>.

Text added:  (p. 2, Supporting information) When the MDR spectral position <...> measured intensity increases <...>

Reviewer 3:

“Check the supplementary information for some inconsistencies ("Fig. ??" on page 3, "other works concerning all-dielectric nanophotonics []" on page 5).

Our response:

We thank the reviewer for pointing out these issues. We have corrected the notation of the pictures and added the necessary link. The entire text was also checked for other typographical errors. Changes are marked in red.

Text added:  (for example, p. 4, section 1: QDs PL analysis) Two types of systems <...> Fig. S4(a), <...> Fig. S4(d).

Reviewer 4 Report

The manuscript describes the experimental and computational investigation of GaAs nanopillar oligomers driven by collective magnetic modes which shows that the excitation of magnetic Mie-type resonance by linearly polarized light in the nanopillar with embedded InAs quantum dots leads to quantum emitters absorption efficiency enhancement.

The topic is novel, the methods are appropriate for achieving the objectives. The results may represent significant advancement in nanoscale single-photon source development.

The experimental part is well planned and executed with excellent results. However, the numerical simulations and modelling are a bit weak, with relatively low reproducibility because simulation details have not been introduced in section Materials and method. For both experimental and computational methods, any commercial products/software and instrument used must be accompanied with product information (serial No., brand, company, city, country).

In conclusion, a major revision is needed to address these shortcomings before further consideration in Nanomaterials.

More details for consideration:

1. both nanopillar and nanocylinder have been used which may cause confusion.

2. The authors stated in Conclusion, “InAs QDs PL enhancement with MDR of GaAs oligomer has been numerically predicted and experimentally realized.” However, the narrative of the manuscript and ordering of materials & methods and results do not reflect this predictive relation.

3. The authors’ effort and discovery are in line with previous investigations. Please also cite DOI: 10.1021/acs.jpcc.5b09751

4. What is the difference between Fig. 3d and Fig. S4?

5. SI appears not have been finalized because it contains many “??”.

6. Fig. S8, caption should introduce left panel before right.

7. Is it possible to plot the difference between experimental and simulated cross-section spectra?  

Author Response

We are thankful to all reviewers for careful reading of the manuscript and pointing out the issues and inaccuracies in it.

We thank Reviewer #4 for his/her detailed analysis of the manuscript and pointing out weak points in numerical results representation. Below we provide responses to his/her remarks.

Reviewer 4:

“For both experimental and computational methods, any commercial products/software and instrument used must be accompanied with product information (serial No., brand, company, city, country).

Our response:

We thank the reviewer for his/her comment and add additional information to the text.

Text added:  (p.3, main text) <...> the halogen lamp with lenses~(L1) being part of the lamp (Thorlabs - SLS201L/M)) <...> the collimator (C), doublet Thorlabs AC254-60-B-ML, <...> the lens~(L2), doublet Thorlabs AC254-150-B-M and the objective lens~(O) Lomo 20x etc

Text added:  (p.4, main text) <...> in Lumerical FDTD software.

Text added:  (p.8, Supporting Information ) <...> in Lumerical FDTD software. The geometry <...>

Reviewer 4:

“both nanopillar and nanocylinder have been used which may cause confusion.

Our response:

We thank the reviewer for his/her comment. We have replaced nanocylinders in the main text and supporting  information with nanopillars, as it is a more appropriate designation.

Text added:  (for example, p. 2, Introduction section) In this work, we experimentally and numerically <...> nanopillars

Reviewer 4:

“The authors stated in Conclusion, “InAs QDs PL enhancement with MDR of GaAs oligomer has been numerically predicted and experimentally realized.” However, the narrative of the manuscript and ordering of materials & methods and results do not reflect this predictive relation.

Our response:

We thank the reviewer for this valuable comment. We rewrote the statement in a form that is more consistent with the logic of the work.

Text added:  (p. 7, Summary section) To summarize, InAs QDs PL enhancement with MDR of GaAs oligomer has been experimentally realized. Physical principles underlying the gain were discussed and numerical calculations were carried out to supplement the achieved results.

Reviewer 4:

“The authors’ effort and discovery are in line with previous investigations. Please also cite DOI: 10.1021/acs.jpcc.5b09751

Our response:

We studied the paper proposed by the referee for citing. While the problem studied in this paper is very important, it is not directly related to the presented work. However, while the CdSe QDs is one of the potential candidates for manufacturing a single-photon source, we included this link (together with additional one) to the introduction section.

Text added:  (p. 2, paragraph 1, main text) Interaction of QDs with various semiconductor [33–38] and <...>.

Reviewer 4:

“What is the difference between Fig. 3d and Fig. S4?

Our response:

We thank the reviewer for raising this question. These figures are the same. The pictures were duplicated for simplicity, however, as we understand, it confuses the reader. We removed Fig. S4 from supporting information and refer to the main text. We also changed the numbering of pictures in the main text.

Reviewer 4:

“SI appears not have been finalized because it contains many “??”.

Our response:

We thank the reviewer for this comment. We carefully reviewed the text and corrected inaccuracies.

 Text added:  (for example, p. 4, section 1: QDs PL analysis) Two types of systems <...> Fig. S4(a), <...> Fig. S4(d).

Reviewer 4:

“Fig. S8, caption should introduce left panel before right.

Our response:

We thank the reviewer for his/her comment. We have changed the caption for this figure.

 Text added:  ( p. 7, Fig. S7)

Reviewer 4:

Is it possible to plot the difference between experimental and simulated cross-section spectra?

Our response:

We thank the reviewer for this advice. The finite element method in the COMSOL Multiphysics software package was used to calculate the scattering spectra of quadrumers with diameters ranging from 210 nm to 290 nm and a constant height of 400 nm and distance between nanoparticles of 100 nm. In the simulation, pumping in the form of a plane wave was used. We superimposed the numerical scattering cross-sections on the experimental results. The results are presented in Fig. S10 in Supporting Information for the resonant quadrumer. The diameter in calculations was 220 nm, which coincides with the parameters of the real oligomer. Appropriate comments about calculations were added to the main text and supporting information.

 Text added:  ( Supporting Information p. 8) We also calculated the emission spectra <...>

 Text added:  (Main text  p. 4) Numerical calculations of the scattering <...>

Round 2

Reviewer 4 Report

The authors have addressed all the comments and it is now ready for publication.